# A New Diagnostic Approach for Periprosthetic Acetabular Fractures Based on 3D Modeling: A Study Protocol

**DOI:** 10.3390/diagnostics10010015

**Published:** 2019-12-29

**Authors:** Giuseppe Marongiu, Roberto Prost, Antonio Capone

**Affiliations:** 1Orthopaedic and Trauma Clinic, Department of Surgical Sciences, Cagliari State University, 09124 Cagliari, Italy; anto.capone@tiscali.it; 2Radiology Unit, Azienda Ospedaliera Brotzu, 09134 Cagliari, Italy; roberto.prost@aob.it

**Keywords:** 3D printing, 3D modeling, total hip arthroplasty, revision surgery, periprosthetic acetabular fracture

## Abstract

Periprosthetic acetabular fractures after total hip arthroplasty (THA) are mostly related to low energy trauma reduced bone quality. CT-scan is widely used to evaluate acetabular fractures, however, metal artifacts produced prosthetic implants limit the visualization of the articular surface and bone loss assessment. 3D modeling software allows us to creating tridimensional images of the bony surface, removing the metallic implants trough image segmentation. We highlight the use of 3D modeling and rapid prototyping (3D printing) for the diagnostic process of periprosthetic acetabular fracture around THA. 3D modeling software was used to improve the assessment of fracture morphology and bone quality. Moreover, the 3D images were printed in a real-life size model and used for preoperative implant templating, sizing and surgical simulation.

## 1. Introduction

Periprosthetic acetabular fractures represent an uncommon but challenging complication of total hip arthroplasty (THA), mostly related to traumatic events and pathologic underlining conditions that reduce the structural integrity of supporting bone [1]. Such fractures represent 3.8% of hip revisions, but the number will grow due to a progressive increase in primary hip implants [1]. Therefore, particularly in elderly and fragile patients, these fractures are associated with aseptic loosening, periprosthetic osteolysis and severe bone loss [2].

Standard radiographs alone are not suitable to reliably detect the residual stability of the implant and measure the extent of the fracture and pelvic bone loss [3,4]. Therefore, CT scan is considered the gold standard to define the fracture pattern, in particular with 3D reconstructions providing excellent reconstruction of the fracture [5]. Specific metal artifact reduction (MAR) protocols could improve the quality of the images, however [6], the presence of the prosthetic implants in situ limits the full view of the articular surface and bone loss [7].

3D modeling allows of creating a 3D representation of surfaces or objects from a 2D slice images, such as CT and MRI scans. In case of periprosthetic acetabular fractures, 3D modeling medical software allows us to obtain precise tridimensional reconstructions of the bony surface, virtually removing the metallic implants trough image segmentation [8]. Additional tools consent to analyze the fracture extension, volumetric measurement of the remaining bone stock and the evaluation of implant bone interface [9]. The final phase of 3D modeling is represented by rapid prototyping, in which 3D models can be printed in real-life size plastic models. Several authors reported useful application of 3D printed models for preoperative implant templating, sizing and surgical simulation of hip surgery [10,11]. Clinical application of 3D digital and printed models has been focused particularly on hip arthroplasty for the development of patient-specific instrumentation (PSI) and custom-made implants [8,12,13]. Other reports described the use of 3D models for the preoperative surgical planning of complex pelvic and acetabular fractures [14,15].

In the current report we present our application of 3D modeling and 3D printing techniques in the diagnostic and planning process of periprosthetic acetabular fractures around THA.

## 2. Methods and Results

Life-size 3D models were manufactured using 3D modeling technology and rapid prototyping from the computed tomography (CT) scans of a periprosthetic acetabular fracture waiting for revision THRs. The patient provided signed, informed consent. The study protocol was approved by our institutional review board (Department of Surgical Sciences, University of Cagliari, CA:11_102017, 1 September 2018).

### 2.1. Case Report

A 75-year-old female presented to the emergency department in September 2018, with sudden right groin pain and a limp, after a domestic trauma from the standing position. She had the history of right hip severe coxarthrosis treated with cementless THA in 1998. Implant characteristic were: a porous-coated acetabular shell, a hydroxyapatite-coated titanium femoral stem with a polyethylene on metal bearing (Osteonics^®^ Omnifit^®^, Stryker, Allendale, NJ, USA). The patient did not previously complain any hip pain or discomfort around the hip joint. The analysis of previous radiographs of the pelvis and the right hip showed no signs of aseptic loosening and mobilization of the acetabular cup. The patient underwent anterior–posterior (AP) radiographs of the pelvis and a cross-leg view of the hip (Figure 1a,b). The cup was stable, with moderate signs of periacetabular osteolysis, and no evidence of polyethylene wear. The femoral stem was well fixed without any sign of loosening. No fracture around the acetabulum and the stem were evident.

### 2.2. Computed Tomography with Metal Artefacts Reduction (MAR)

The patient underwent a CT scan of the pelvis. In supine position, patient’s legs are mutually aligned secured to prevent motion with the foot perpendicular to the table with toes pointing straight up. Patient is checked for un-natural tilt or lift of the pelvis. CT parameters on the GE Optima 580 W (General Electric Healthcare, Chicago, IL, USA) were kept constant: Helical scan, 120 kv, 0.5 sec of rotation time, Pitch 1.375:1. Slice thickness was set to 1 mm and spacing at 0.8 mm. Scan starts at the top of the iliac crest and stops at mid-femur or at least 3 cm below existing femoral implant. Field of view (FOV) is 38–44 cm depending on patient size. Optimal Cubic Voxel is set at 0.5 × 0.5 × 1 mm^3^. Specific MAR algorithm, Smart Metal Artifact Reduction software (General Electric Healthcare, Chicago, IL, USA) was used to reduce noise and improve the image quality.

The CT scan revealed a fracture of the posterior wall of the acetabulum with an apparently non-displaced acetabular cup (Figure 2).

### 2.3. 3D Modeling and Rapid Prototyping Process

3D modeling allows of creating a 3D representation of surfaces or volumes from a 2D slice images, such as CT and MRI scans. We, therefore, used a medical image processing software (Mimics Innovation Suite 14.12, Materialise, Leuven, Belgium) to assess the fracture morphology and bone quality, creating a tridimensional digital model of the pelvis, based on the CT scan. Raw DICOM data were imported into the 3D modeling software. Then, image segmentation was performed to differentiate bone from prosthetic implants and the surrounding soft tissue. Segmentation is a generic term for grouping all voxels together in a set that defines a specific anatomic feature or tissue. The first phase is thresholding, which groups all voxels whose density is within a specified range of Hounsfield Unit (HU) values. We used a mask with a HU range from 130 to 1750 in order to exclude metallic and ceramic implants and include both cancellous and cortical bone (Table 1).

Choosing a lower limit of 130 HU, allowed us to include in the 3D model all the cancellous bone, which can have HU values < 200. On the other hand, this wide HU range mask did not completely exclude soft tissues, therefore additional manual segmentation is needed. The final segmentation, with the removal of soft tissues and artifacts, was manually performed using additional tools of the software (region growing, edit mask and crop mask; Figure 3a,b). Eventually, both femurs and metal implants were eventually digitally removed from the corresponding pelvis and a 3D image of the isolated region of interest (ROI) was created. Moreover, the software allows us to calculate the overall bone thickness and cortical thickness of the different regions of the acetabulum. Therefore, a bone quality map was built with a color gradient from red (inferior) to green (excellent; Figure 3c,d). Further analysis included measures of shape, area and spatial location of the fracture and the acetabular bone loss and the center of rotation, compared to the contralateral acetabulum (Figure 3e,f).

Then, 3D images of the ROI were exported in a pdf file format, which allows the operators to evaluate (i.e., zooming, panning, rotate, cross section and measure) the 3D digital model in all the most common type of computer machines. Moreover, a STL. (stereolithographic) file format was transmitted to a 3D printer (Form 3L, Formlabs) for the realization of a life-size model of the entire pelvis (Figure 4).

### 2.4. Classification, Planning and Surgical Procedure

According to the analysis of the 3D images the posterior wall fracture was re-classified as an incomplete posterior column and medial wall acetabular fracture (Figure 3). The acetabular cup appeared slightly migrated medially and cranially (8 mm of cranial migration). The fracture was classified on the digital 3D model as a “spontaneous fracture with less than 50% of bone stock loss” according to Della Valle and Paprosky [18]. The treatment strategy was chosen according to the algorithm proposed by Simon et al. [19], which suggest the acetabular revision surgery bridging or distracting the fracture, without fracture fixation, in case of osteolytic based acetabular fracture with poor bone quality. Preoperative templating was performed on the 3D printed model. The fracture line was incomplete, did not reach the anterior wall and was substantially non-displaced. After a mild reaming of the acetabular cavity, the medial wall bone loss was filled with a patch (75 mm × 75 mm × 5 mm) of synthetic bone substitute (RegenOss, FinCeramica Faenza S.p.A.) and a cementless acetabular cup with three iliac flanges and a caudal hook, manufactured in trabecular titanium, was press fitted in the acetabulum (Delta Revision TT, Lima), augmented trough four additional iliac screws. An ultra-high molecular weight polyethylene acetabular liner and ceramic head and 32-mm BIOLOX delta alumina ceramic femoral head, were inserted.

### 2.5. Outcome

Postoperative AP radiographs of the pelvis and the right hip, showed a well-positioned and fixed implant (Figure 5a).

The patient had a regular postoperative hospital stay, with no need for blood transfusion and no complication recorded. Early passive mobilization in bed was prescribed for the first 30 days. Then the patient achieved the standing position with full weight bearing and the use of two crutches. Moreover, after 3 months from the surgery, the patient underwent to a CT scan of the pelvis, which showed the complete healing of the fracture, with callus formation, and the bone integration of the trabecular cup (Figure 5b,c). The DICOM images were used to build a 3D digital model, which confirmed the CT scan findings (Figure 6). At 12 months after the surgery, the Harris Hip Score was 78 points and the Western Ontario and McMaster Universities Osteoarthritis Index (WOMAC) score was 81 points [20,21,22].

## 3. Discussion

The treatment of periprosthetic acetabular fractures represents a challenging issue for the orthopedic surgeon who has to address both the need for fracture reduction and stability, and an effective joint reconstruction [19,23]. Periprosthetic acetabular fractures often occur in patients in which poor bone quality is often concomitant with periprosthetic bone lysis [2]. CT scan, with MAR protocol, represents the preferred diagnostic method for the evaluation of periprosthetic acetabular fractures and the quantification of periprosthetic bone loss [6].

However, new 3D modeling software and their application in orthopedics, as diagnostic tools, demonstrated several useful applications [15]. 3D modeling allows of creating a 3D representation of surfaces or volumes from a 2D slice images, such as CT and MRI scans. The 3D models can be then printed into plastic 3D real-size models trough the so-called “rapid prototyping”. A 3D printed model provides visual and tactile reproduction of the fracture and bony anatomy. This enables an enhanced understanding of the anatomy and facilitates enhanced preoperative planning [11]. Several authors reported the benefits of 3D modeling and 3D printed aids for preoperative planning and templating of complex acetabular fractures [24,25,26]. Maini et al. [25] evaluated the outcome patients with acetabular fractures, using a 3D modeling software and a 3D printer to create patient specific (PSI) templates over which 3.5-mm reconstruction plates were manually contoured preoperatively and used for fixation. The results, compared with a non-PSI group of patients, indicated that the patients of the PSI group underwent to shorter time of surgery, less blood loss and better anatomical reduction. Li et al. [26] found shorter mean operative time (−43 min) and less complications when 3D modeling was used for planning and templating acetabular fractures.

Another important advantage of 3D modeling software is the possibility of virtually removing the metallic implants allowing the direct view of fracture lines or bone defects underlying the metal cup. This is possible trough image segmentation, which is differentiation of different tissues and materials bone, like prosthetic implants and the surrounding soft and bony tissue, according to similar radiological characteristics [11]. This software provides different segmentation features, which can be both automatically and manually performed, in order to accurately remove residual soft tissues and artifacts. A few authors applied 3D modeling techniques for the planning of acetabular revision arthroplasty [27,28]. Kavalerskji et al. reported a case series of 17 hip revisions for complex acetabular defects in which 3D models were used for bone defect classification and pre-operative cup and augment templating. They similarly found almost perfect correspondence between pre-operative and intraoperative findings [27]. Other authors, 3D modeling software for preoperative planning of revision arthroplasty and the realization of 3D printed custom made 3-flanged cups [29].

For the best of our knowledge, the use of 3D modeling techniques for periprosthetic acetabular fractures has not been yet described. In the case we presented, 3D modeling software had a crucial role in decision making for the treatment of periprosthetic acetabular fractures. Firstly, the digital model consented to explore the articular anatomy and bony surface without the metal cup, revealing the exact location, extent and direction of the fracture, and changing the initial classification made according to CT scan images alone. Then the analysis feature of bone quality and thickness suggested that the underlying fragility and osteolytic pattern as the cause of the fracture. Classified as a spontaneous fragility fracture with substantial bone stock loss [18], the patient therefore underwent acetabular revision surgery without fracture fixation, as proposed by Simon et al. [19]. This solution has been reported by other authors with good results in terms of fracture healing and functional results [30,31].

Preoperative screw position simulation allowed for the best use of available bone stock and helped ensure the best construct stability. The trajectory simulation was carried out on the 3D models, allowing for improved accuracy orienting them towards areas of good quality bone according to the “bone quality map”. Moreover, preoperative templating performed on the 3D printed model, was particularly helpful for deciding the shape of the superior flanges of the acetabular cup, which have to be placed at strict contact to the iliac bone.

As reported by previous studies [27,29], during the surgical procedure, we found a full correspondence with the implant size, the screws measure preoperatively planned. We think that the different phase of segmentation are crucial for determine the real amount of bony tissue, which is removed from the model, building a personalized model. Therefore, the a wide HU values range for thresholding and then the use of manual segmentation tools are crucial points for obtaining an accurate representation of reality. This probably led us to a safer surgery and shorter operative time. Moreover we obtained a rapid bone integration of the implant and fracture healing, as revealed in the CT scan at two months after the surgery.

## 4. Conclusions

This study highlights the application of 3D modeling software and 3D printing could be a helpful additional tool for diagnosing, decision making and treatment of periprosthetic acetabular fractures. The use of 3D modeling software showed that periprosthetic acetabular fractures can be better addressed, compared to plain radiograph and CT scans, providing additional measurement tools, which allow the volumetric analysis of bone defects and bone quality assessment. Digital and 3D printed models can be used for preoperative surgical planning and templating. Further investigation, with a large number of patients, have to confirm whether this protocol and novel diagnostics tools can improve functional outcomes and allow the development of simpler treatment algorithm. Moreover, future efforts should be focused into the improvement mage artifacts reduction and into understanding the role of bio-printing technologies development for biological reconstruction of bone defects.

## Figures and Tables

**Figure 1 diagnostics-10-00015-f001:**
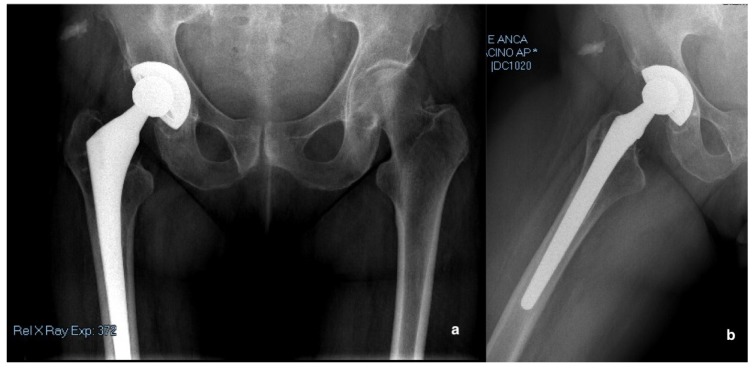
(**a**) Anterior–posterior (AP) pelvis and (**b**) lateral view of the right hip radiographs showed mild signs of periacetabular osteolysis without evidence of implant loosening and acetabular fracture.

**Figure 2 diagnostics-10-00015-f002:**
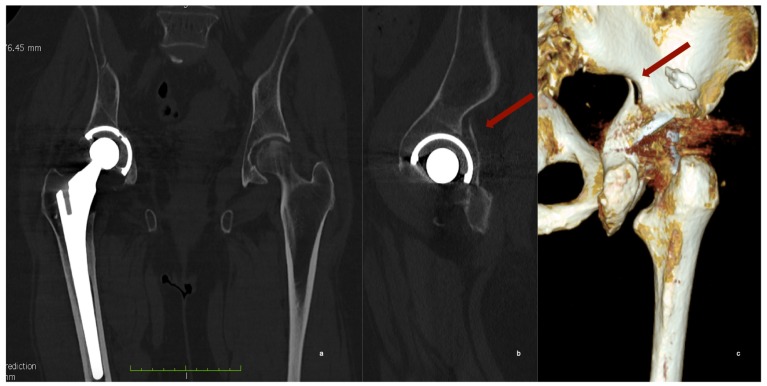
CT scan of the pelvis. (**a**) Coronal view shows a slightly medially protruded acetabular cup; (**b**) the sagittal view of the hip, revealed the posterior wall fracture of the acetabulum and (**c**) in the tridimensional reconstruction the fracture is clearly visible, but its extension is hidden by image artifacts.

**Figure 3 diagnostics-10-00015-f003:**
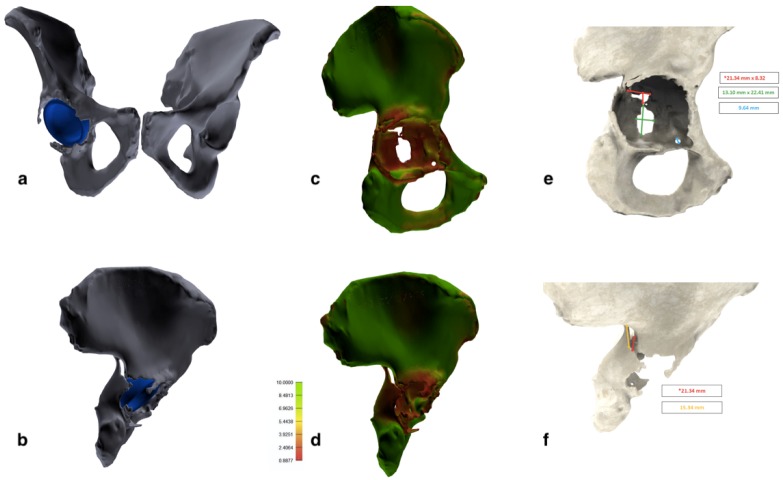
Tridimensional images elaborated with the 3D modeling software. (**a**,**b**) Entire pelvis with the acetabular cup retained. Femurs and femoral stem were removed during segmentation. (**c**,**d**) Bone quality map shows regions with normal bone quality (green) and regions with low bone quality and thickness (red). (**e**,**f**) Measurements of the bone defect area and fracture extension.

**Figure 4 diagnostics-10-00015-f004:**
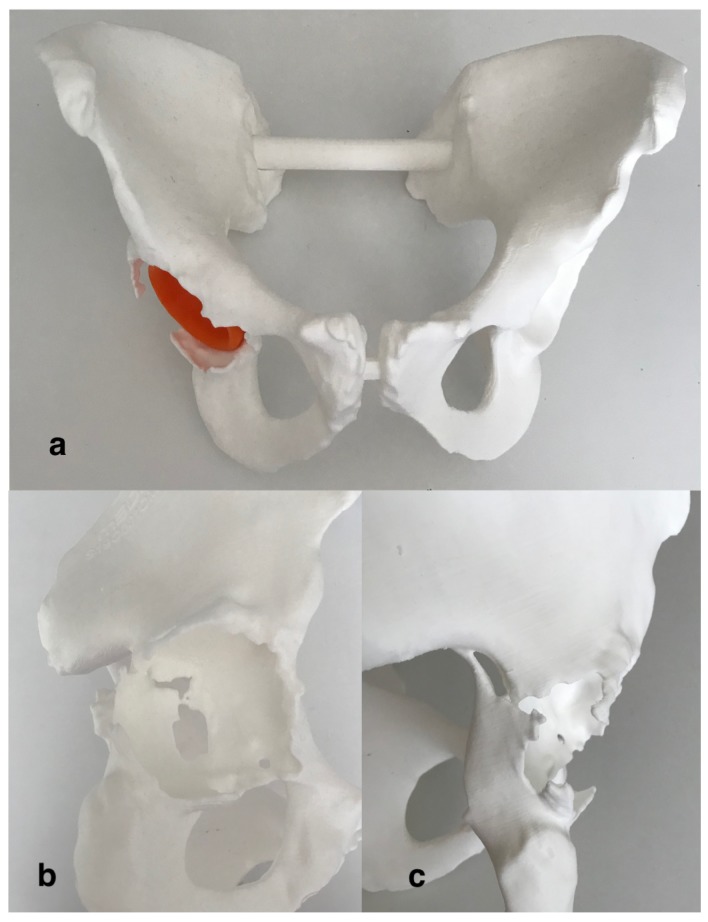
(**a**) 3D printed, real-size, plastic model of the entire pelvis. (**b**,**c**) Particular of the fracture of the medial wall and posterior column.

**Figure 5 diagnostics-10-00015-f005:**
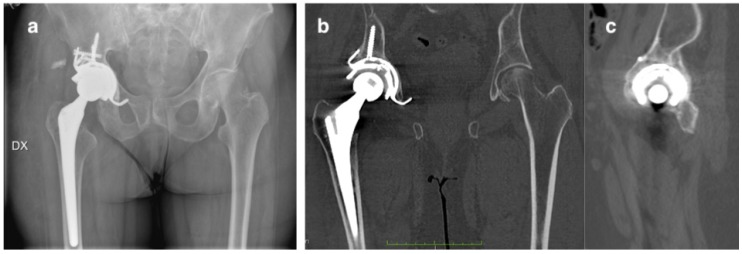
(**a**) Postoperative AP radiograph of the pelvis and (**b**,**c**) CT scan of the pelvis at 3 months after the surgery shows the good implant positioning and the complete fracture healing with abundant callus formation.

**Figure 6 diagnostics-10-00015-f006:**
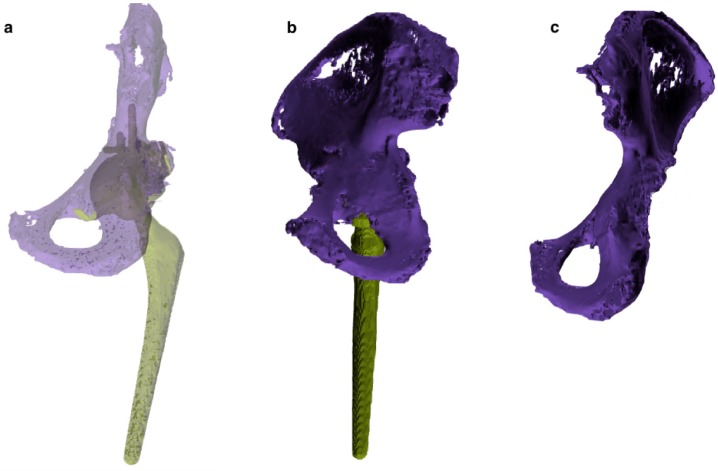
3D modeling digital reconstruction. The posterior column and medial wall of the acetabulum have been restored.

**Table 1 diagnostics-10-00015-t001:** Mechanical properties of materials involved in the CT scan analysis and segmentation of Case 1 [16,17].

Components	Materials	Elastic Modulus (MPa)	Density	Hounsfield Unit (HU) Values	
Cortical bone		Cortical bone	17,300	1.6–2 g/mm^3^ (porosity 10%)	1500
Cancellous bone		Cancellous bone	400	0.05–0.3 g/mm^3^ (porosity 70%)	200–700
Femoral prosthesis (Osteonics, Omnifit, Stryker)		Titanium alloy + Hidroxyapatite	110,600	4.43 g/cm^3^	3071
Acetabular cup (Osteonics, Omnifit, Stryker)		Titanium alloy + Hidroxyapatite	110,600	4.43 g/cm^3^	3071
Acetabular revision cup (Delta Revision TT, Lima)		Trabecular Titanium	8963	4.43 g/cm^3^ (porosity 60%)	2840–3071
Screws		Titanium alloy	110,600	4.43 g/cm^3^	3071
Polyethilene insert		Ultra-High-Molecular-Weight Polyethylene	1500	0.93 g/cm^3^	19–53
Metallic femoral head		Cobalt–chromium alloy	230,000	8.2 g/cm^3^	3071
Ceramic femoral head		Alumina	350,000	4.1 g/cm^3^	3071

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
