# Peer review of "A New Diagnostic Approach for Periprosthetic Acetabular Fractures Based on 3D Modeling: A Study Protocol"

_diagnostics, 2019, doi:10.3390/diagnostics10010015_

Round 1
Reviewer 1 Report
The authors do not include post OP data. How was the patients evaluation after surgery. where the complications? Were there follow up issues?
Author Response
Thank you for your comments.
We included the post operative evaluation in line 131 - 138 of the original manuscript. The follow up at the time of the last evaluation was 12 months.
"The patient had a regular postoperative hospital stay, with no need for blood transfusion and no complication recorded. Early passive mobilization in bed was prescribed for the first 30 days. Then the patient achieved the standing position with full weight-bearing and the use of two crutches. Moreover, after 3 months from the surgery, the patient underwent to a CT scan of the pelvis which showed the complete healing of the fracture, with callus formation, and the bone integration of the trabecular cup (figure 5b, 5c). The DICOM images were used to build a 3D digital model which show confirmed the CT scan findings (figure 6). At 12 months after the surgery, the Harris Hip Score was 78 points and the WOMAC score was 81 points"
Reviewer 2 Report
Overall, this is a well written paper. However, I have the following questions:
1) Is there a need to get consent from the patient to publish this study?
2) Can the authors provide more details on the procedure used in MIMICS? In particular on this particular statement in the manuscript: "A specific feature of the software allow to build a bone quality map...".
3) It would be useful to repeat this study multiple times to validate the findings.
Author Response
Thank you for your precious comments:
1) Is there a need to get consent from the patient to publish this study?
Yes, we obtained the consent from the patient (please see the attachment)
2) Can the authors provide more details on the procedure used in MIMICS? In particular on this particular statement in the manuscript: "A specific feature of the software allow to build a bone quality map...".
we have modified the text in line 91 - 93
The software allow to calculate the overall bone thickness and cortical thickness of the different regions of the acetabulum. Therefore an image showing the bone quality was build with a colour gradient from red (inferior) to green (excellent).
3) It would be useful to repeat this study multiple times to validate the findings.
Yes, of course. We are now improving the methodology and we have applied already on one new case. However, the patient did not agree for publication.

Reviewer 3 Report
This case report represent a great advance in surgery and it is very interesting to be considered to publish in this journal. I have several issues to be clarified before to finish the process of review:
Page 1 line 31 "multiple studies" but just one reference. It is important to add more references here. In my opinion, Metal artefact reduction MAR protocols should be detailed and the bone loss should be determined also in quality. 3D modeling software needs a paragraph with a comment about image acquisition and finite element methods. Stiffness is not commented in the introduction and it is relevant it should be mentioned also in the discussion. A brief paragraph about the epidemiology of this affection is omitted. Being a case report may it should be contextualized in terms of incidence in the population. It seems that this procedure can be adapted to other muscoeskeletal pathologies, what is the scope? D modeling is a subsection but it is not explained. 3D modeling should be described, why and how is used with its physical meaning. The material is commented as a commercial purchase but is important to detail in a table its properties, I mean, stiffness, density,... at the end is 3D personalized printing, the procedure is also important to be developed. The harris Hip Score and WOMAC score should be cited. A future protocol should be suggested in the discussion. The role of 3D printing and bioprinting is interesting to be mentioned in the manuscript also with a future work or tendencies in the conclusion.
Author Response
Dear reviewer, we really appreciated your precise and accurate comments.
Here you can find my point by point response.
1) Page 1 line 31 "multiple studies" but just one reference. It is important to add more references here.
According to your suggestion, we have updated our reference list.
2) In my opinion, Metal artefact reduction MAR protocols should be detailed and the bone loss should be determined also in quality.
we changed the text and included MAR protocol characteristics (line 70 - 88)
3) 3D modeling software needs a paragraph with a comment about image acquisition and finite element methods.
we changed the text according to your suggestion (line 35 - 41 )
4) Stiffness is not commented in the introduction and it is relevant it should be mentioned also in the discussion.
we believe that stiffness is not an essential parameter in this case, because we are not executing a finite element analysis ( i.e. strength, torsion), but we are using this CAD software for diagnostic purpose. Therefore it could more helpful to mention the density of the material involved and the Hounsfield UNit values needed to accurately differentiate tissues from metals (see table 1 in the file attached.)
5) A brief paragraph about the epidemiology of this affection is omitted. Being a case report may it should be contextualized in terms of incidence in the population.
we agree with your point. we provided the requested informations. (see line 23 - 26)
6) It seems that this procedure can be adapted to other musculoskeletal pathologies, what is the scope?
The main point of orthopedic research efforts in now oriented to improve the evaluation of bone stock in patient undergoing to revision surgery of both hip, knee and shoulder. As long as these technologies will become over and over more reliable and cheaper, I believe that 3D modeling will become the gold standard for both diagnostic and operative purposes. See the discussion for this point.
7) 3D modeling is a subsection but it is not explained. 3D modeling should be described, why and how is used with its physical meaning.
we agree with your point. We provided the requested information (line 89 - 109)
8) The material is commented as a commercial purchase but is important to detail in a table its properties, I mean, stiffness, density,... at the end is 3D personalized printing, the procedure is also important to be developed.
please see point 4.
9) The Harris Hip Score and WOMAC score should be cited.
we have updated the reference list.
10) A future protocol should be suggested in the discussion. The role of 3D printing and bioprinting is interesting to be mentioned in the manuscript also with a future work or tendencies in the conclusion
we agree with your point. Considering the amount of information we added in the manuscript we changed our study purpose. I believe that now represents not only a case report description but more likely a protocol description. (See conclusion)